# Diagnostic Challenges in Canine Parvovirus 2c in Vaccine Failure Cases

**DOI:** 10.3390/v12090980

**Published:** 2020-09-03

**Authors:** Hiu Ying Esther Yip, Anne Peaston, Lucy Woolford, Shiow Jing Khuu, Georgia Wallace, Rohan Suresh Kumar, Kandarp Patel, Ania Ahani Azari, Malihe Akbarzadeh, Maryam Sharifian, Reza Amanollahi, Razi Jafari Jozani, Aliakbar Khabiri, Farhid Hemmatzadeh

**Affiliations:** 1School of Animal and Veterinary Sciences, The University of Adelaide, Roseworthy Campus 5371, Australia; hiuyingesther.yip@adelaide.edu.au (H.Y.E.Y.); anne.peaston@adelaide.edu.au (A.P.); lucy.woolford@adelaide.edu.au (L.W.); jing95sk@hotmail.com (S.J.K.); georgiawallacedvm@gmail.com (G.W.); a1686484@student.adelaide.edu.au (R.S.K.); kandarp.patel@adelaide.edu.au (K.P.); maryam.sharifian@adelaide.edu.au (M.S.); razijafarijozani@gmail.com (R.J.J.); aliakbar.khabiri@adelaide.edu.au (A.K.); 2Department of Microbiology, Gorgan Branch, Islamic Azad University, Gorgan 4914739975, Iran; ania_783@yahoo.com; 3Faculty of Veterinary Medicine, Shiraz University, Shiraz 69155-71441, Iran; malihe.akbarzade@gmail.com (M.A.); reza.vet93@gmail.com (R.A.)

**Keywords:** Canine Parvovirus Type 2c, Vaccine failure, qPCR, antigen rapid CPV/CCV tests, test sensitivity, test specificity, Bayesian latent class analysis

## Abstract

In this study, three different diagnostic tests for parvovirus were compared with vaccination status and parvovirus genotype in suspected canine parvovirus cases. Faecal samples from vaccinated (N17) and unvaccinated or unknown vaccination status (N41) dogs that had clinical signs of parvovirus infection were tested using three different assays of antigen tests, conventional and quantitative PCR tests. The genotype of each sample was determined by sequencing. In addition to the suspected parvovirus samples, 21 faecal samples from apparently healthy dogs were tested in three diagnostic tests to evaluate the sensitivity and specificity of the tests. The antigen test was positive in 41.2% of vaccinated dogs and 73.2% of unvaccinated diseased dogs. Conventional PCR and qPCR were positive for canine parvovirus (CPV) in 82.4% of vaccinated dogs and 92.7% of unvaccinated dogs. CPV type-2c (CPV-2c) was detected in 82.75% of dogs (12 vaccinated and 36 unvaccinated dogs), CPV-2b was detected in 5.17% dogs (one vaccinated and two unvaccinated) and CPV-2a in 1.72% vaccinated dog. Mean *Ct* values in qPCR for vaccinated dogs were higher than the unvaccinated dogs (*p* = 0.049), suggesting that vaccinated dogs shed less virus, even in clinical forms of CPV. CPV-2c was the dominant subtype infecting dogs in both vaccinated and unvaccinated cases. Faecal antigen testing failed to identify a substantial proportion of CPV-2c infected dogs, likely due to low sensitivity. The faecal samples from apparently healthy dogs (*n* = 21) showed negative results in all three tests. Negative CPV faecal antigen results should be viewed with caution until they are confirmed by molecular methods.

## 1. Introduction

Canine parvovirus (CPV) belongs to the family *Parvoviridae*, the subfamily *Parvovirinae* and the genus *Protoparvovirus* (previously *Parvovirus*). *Protoparvovirus* comprises feline panleukopenia virus (FPV), canine parvovirus (CPV), mink enteritis virus (MEV) and raccoon parvovirus (RPV) [1]. The clinical form of canine parvovirus (CPV) was first reported in the United States in 1978 and the isolated virus was named CPV type-2 (CPV-2) to distinguished from canine parvovirus type 1 (CPV-1) which causes canine minute virus disease [2]. The emergence of CPV-2 in the United States as a canine pathogen was a consequence of “jumping the species barrier” from FPV to CPV-2 [3,4]. After successful species jump, CPV-2 was further selected for growth within canine cells, making dogs a better host [5]. The establishment of CPV-2 in the dog population was strongly associated with post-transmission adaptation or pre-existing FPV variants in domestic or wild carnivores such as mink and foxes [6,7].

After the emergence of CPV-2 in the late 1970s, the virus rapidly spread around the world, causing fatal enteric and myocardial disease in dogs [8,9]. Serological studies in late 1978 confirmed the presence of CPV-2 antibodies in dogs in Japan, Australia, New Zealand, and the United States, demonstrating that the virus had spread around the world in less than 6 months [10]. CPV-2 virus was then rapidly replaced worldwide by an antigenically and genetically distinct strain of canine parvovirus 2a (CPV-2a) [11] and, in early 1980, CPV-2a was recognized as the principal parvovirus found in dogs [12,13]. Due to very high genomic substitution rates in CPV-2 isolates, novel antigenic variants of the virus have continued to evolve [14]. In 1984, a new variant of the virus known as CPV-2b emerged first in the United States [15], then spread worldwide and caused disease in both dogs and cats [4].

The third variant of the virus, CPV-2c, was first identified in 2000 in Italy [16,17], although retrospective studies showed it had been circulating in Germany for 4 years previously [18]. It spread briskly across Europe [19,20], America [17,21], Asia [22,23] and Australia [24]. Field studies showed that CPV-2c is becoming the predominant variant affecting the dog population worldwide [25,26,27]. The major antigenic difference between CPV-2 variants is a key amino acid at residue 426 in epitope A of the Viral Protein 2 (VP2). At this residue, CPV-2a has asparagine, CPV-2b has aspartic acid and CPV-2c has glutamic acid [28,29,30]. The contributions of genetic and antigenic variations in the VP2 antigen are not well understood and are thought to support cross-species transmission and possibly the adaptation of the virus to the host via genetic selection, but controlled studies of the relative pathogenicity of the different strains are lacking [1,3,4,11,14]. In addition, residues elsewhere contribute to viral pathogenicity, vaccine failure, virus host range and the sensitivity of diagnostic tests [1,18,31]. Decaro (2008, 2009) and Sehata (2017) have discussed the role of the alteration of viral surface proteins and its contributions to vaccine failure and the poor performance of the diagnostic tests in clinical forms of CPV-2c in vaccinated dogs [31,32,33].

Vaccination with modified live CPV is the main method of controlling CPV disease and increasing population immunity in dog environments [10]. Live attenuated parvovirus vaccines are safe and provide protective immunity when administered at appropriate intervals [18,34]. A major cause of vaccine failure is thought to be the administration of the final vaccine dose to puppies less than 16 weeks old, when maternal antibodies can interfere with the development of immunity, although many other causes are possible [24,35].

To date, the available vaccines for CPV are derived from CPV-2 or CPV-2b strains only and it remains unresolved as to what extent vaccination failures are associated with suboptimal protection against CPV-2c [36]. Decaro and Buonavoglia (2012) have highlighted the poor sensitivity of the antigen detection tests and high CPV antibody titres in the gut lumen. The mucosal antibodies block the viral particles and cause negative results in immunochromatographic tests, even in clinical cases of CPV [1].

To investigate this, rapid, specific and sensitive tests for different virus strains would be ideal tools to identify shed virus in the faeces of infected animals. Quantitative real-time PCR (qPCR) methods have shown high sensitivity in the detection of dogs shedding low quantities of CPV, detecting as few as 10 copies of virus DNA [37]. The main objectives of this study are to genotype CPV-2 isolates of clinical cases of canine parvovirus infections and compare the performance of a novel qPCR test with a commercial rapid in-clinic test for the detection of CPV genotypes in clinical cases.

## 2. Materials and Methods

### 2.1. Samples

A total of 58 faecal samples from diseased dogs in different states of Australia were submitted to the virology laboratory in the School of Animal and Veterinary Sciences, University of Adelaide, between February 2015 and February 2019. All of the samples were from dogs with mild to severe diarrhoea and/or other clinical signs suggestive of CPV. To avoid any bias from the interfering maternal antibodies with the vaccination history, the samples with the age range of 9 to 16 weeks were removed from this study. In addition to faecal samples from diseased dogs, 21 more samples were collected from apparently healthy dogs as negative control with the previous history of CPV vaccination not earlier than two months. The vaccination status and geographical distribution of the samples are shown in Appendix A.

### 2.2. Antigen Detection Test and Molecular Methods

The faecal samples were initially tested for parvovirus antigens using the Antigen Rapid CPV/CCV Ag kit according to the manufacturer’s instructions (Bionote, Geyonggi, Korea).

Total DNA was extracted from 200 mg of fresh or frozen faecal samples using the QIAamp^®^ DNA Stool kit (QIAGEN Inc., Valencia, CA, USA). The final volume of DNA for each sample was 100 µL. To screen the samples, conventional PCR was carried out on all of the samples using primers, as previously described [24]. In brief, a master mix was made up with 8.84 µL distilled DNase free water, 5 µL AllTaq^®^ (QIAGEN), 4 µL of each DNA sample or distilled water for negative control and 0.8 µM each of the forward primer F4(9) (CATACATGGCAAACAAATAGAGCATTGGGC) and reverse primer R4(9) (ATTAGTATAGTTAATTCCTGTTTTACCTCC). The thermal protocol started at 95 °C for 2 min, then 40 cycles of 95 °C for 5 s, 50 °C for 15 s and extension at 72 °C for 10 s.

To determine the CPV genotypes, positive PCR products were purified and sequenced in both forward and reverse directions using above-mentioned primers. The VP2 sequences were edited and assembled using BioEdit package v.7.0.4.1 [38]. To determine relationships among clinical case-derived sequences and known CPV-2 genomes, ClustalW and MEGA (Version 6.06) programs were used for alignment and phylogenetic analysis of nucleotide sequences from CPV-2 genomes retrieved from GenBank and the sequences obtained from clinical cases [39,40].

### 2.3. Quantitative PCR for Detection of Virus Load

To measure CPV viral load, a real-time qPCR assay was developed using QuantiFast SYBR Green PCR (QIAGEN Inc., Valencia, CA, USA), forward primer AGCTACTATTATGAGAACCAGCTGAG and reverse primer CCTGCTGCAATAGGTGTTTTAA. The primers were designed to amplify a 98-bp fragment from position 981 to 1079 of the conserved regions of VP2 gene of CPVs available in GenBank. The assays were performed using Eco™ Real-Time PCR System (Illumina, San Diego, CA, USA). The reaction contained 10 µL of 2X QuantiFast SYBR Green Master Mix, 1 µM of each primer and 5 µL of faecal sample DNA in a 20 µL PCR reaction. The thermal profile for qPCR was 95 °C for 5 min, followed by 40 cycles of 95 °C for 10 s and 60 °C for 30 s, with a melt curve analysis at 60–95 °C with increments of 0.3 °C for 5 s.

To construct a control sample for qPCR, we engineered the pGEM^®^-T Easy plasmid vector (Promega, Madison, WI, USA) to contain the 98-bp VP2 fragment described above. The plasmid copy number per microliter was calculated using an online calculator (http://cels.uri.edu/gsc/cndna.html).

Reproducibility of the assay was determined by replicating the assay for each sample on two different days for a total of 3 replicates. To set up a standard curve, the purified plasmid was subjected to 10-fold serial dilutions from 10^−3^ to 10^−13^, representing 8.36 × 10^7^ down to 8.36 copies of CPV viral DNA per microliter. One microliter of each dilution was used as the template in qPCR reactions. The standard curve was constructed by plotting the logarithmic values of serial dilutions of the plasmid preparation versus the threshold cycle (*Ct*) value in the qPCR reaction. The amplification data were collected and analysed using Eco™ Real-Time PCR software (Illumina, CA, USA). A sample was considered positive for CPV amplification when the mean *Ct* value of three replicates was less than the mean *Ct* value for negative controls plus 2 times the standard deviation of the same replicates (cut-off).

To compare the limit of detection of the antigen detection test with qPCR, five faecal samples with high virus loads that were tested in qPCR were selected. Tenfold serial dilutions from 1 to 10^−8^ of each faecal sample were made in Phosphate Buffered Saline (PBS) and tested by the antigen detection kit. In parallel, DNA was isolated from each dilution and subject to the abovementioned qPCR method.

### 2.4. Statistical Analysis

The association between the CPV vaccination status of the dogs and the test status (positive or negative) on PCR and antigen detection tests was assessed using Fisher’s exact test. Similarly, one-way ANOVA was performed to assess the difference in mean *Ct* values between vaccinated dogs and dogs with unknown vaccination status. The *Ct* values were log transformed for ANOVA analysis and were back transformed for the reporting of the results. The association between the genotype of a PCR-positive sample, obtained from sequencing analysis and the CPV vaccination status of dogs, was assessed using Fisher’s exact test. A *p*-value of <0.05 was considered to be statistically significant. An analysis for the assessment of associations and one-way ANOVA were performed using the R software (v 3.5.1) [41].

Bayesian latent class (BLC) analysis was used to determine the sensitivity (Se), specificity (Sp), positive predictive value (PPV), and negative predictive value (NPV) of the three tests. The BLC analysis was executed using a latent class model, assuming that none of the tests was considered as a gold standard test. A posterior distribution was derived combining the observed data with prior knowledge about the parameters (Se_qPCR_, Se_AG_, Se_PCR_, Sp_qPCR_, Sp_AG_, and Sp_PCR_) where AG is the rapid antigen test. All three tests were assumed to conditionally depend on the true CPV exposure status and on constant test accuracy in the test samples. Prior information on Se and Sp for qPCR, antigen tests and conventional PCR were available from data in past studies and/or from expert opinion from the laboratory, applying the test for research purposes (Table 1). The conventional PCR was expected to be more sensitive than the rapid antigen detection test and therefore the most likely Se_PCR_ value was assumed to be 90%, which corresponded to a beta distribution with α and β values of 5.38 and 1.49, respectively (Table 1). The priors for the mean of the log-transformed *Ct* values from qPCR for CPV-positive and -negative groups were set to 2.7 and 3.3, respectively. The standard deviation (SD) for the mean of the *Ct* values in the CPV-positive and -negative dog populations were set to 0.15 and 0.3, respectively. Independent beta prior distributions were derived for the parameters using Betabuster software (vetmed.ucdavis.edu/cadms/diagnostic%20tests/software.cfm) to check for their uncertainty using modal or most probable values and 5th and 95th percentiles.

The model estimating Se_qPCR_, Se_AG_, Se_PCR_, Sp_qPCR_, Sp_AG_, and Sp_PCR_ was fitted to the data in WinBUGS [42], assuming independence between the dogs in study population, using the Bayes Continuous Diagnostic Test (BCDT) software package version 3.9 [43]. The BCDT software is based on latent class analysis methods to determine parameters for diagnostic tests with dichotomous and continuous outcomes [43,44]. The *Ct* values from qPCR were transformed on a natural logarithmic scale before being analysed in the BCDT software and then back transformed for the reporting of results. Each model was run for 50,000 iterations with a burn-in discard period of 5000 iterations. The convergence of the posterior iterates was assessed in WinBUGS by examining history trace plots and by specifying two sets of dispersed initial values and examining the convergence statistic. Additionally, the BLC analyses provided positive predictive values (PPV) and negative predictive values (NPV) for all three tests. The influence of the parameter priors on posterior estimates of Se, Sp, PPV, NPV of qPCR, PCR, and antigen detection tests was evaluated in a sensitivity analysis using uniform, optimistic and pessimistic priors. To maximise Se_qPCR_, a cut-off *Ct* value was selected from the two-graph receiver operating characteristic (TG-ROC) dataset generated by WinBUGS. Se_qPCR_, Sp_qPCR_, PPV_qPCR_, and NPV_qPCR_ at the optimised cut-off are reported. The samples were reclassified as positive and negative based on the optimised cut-off.

The Kappa statistic was estimated to check the level of agreement between the antigen detection test and PCR using the ‘vcd’ package (v 1.4-4) [45] in R software. Based on the Kappa statistic, the test agreement can be interpreted as: <0.2 = slight agreement; 0.2–0.4 = fair; 0.4–0.6 = moderate; 0.6–0.8 = substantial; >0.8 = almost perfect [45,46,47]. To compare the virus load in vaccinated and unvaccinated groups, mean *Ct* values and the standard error of the mean for each sample were calculated and compared with the negative samples.

## 3. Results

### Antigenic and Molecular Detection

All dogs included in the study presented with gastrointestinal clinical signs consistent with parvoviral disease. The clinical history of the tested dogs showed that among 58 referred samples, 17 patients had a history of CPV vaccination and 41 patients were unvaccinated or of unknown CPV vaccination status. The age range for the vaccinated dogs was from 5 months to 6.5 years.

Thirty-seven of the 58 dogs (63.79%) were positive for CPV by faecal antigen detection testing (Table 2). The proportion of dogs positive in the faecal antigen detection test were significantly lower in vaccinated dogs compared to dogs with an unknown vaccination status (*p* = 0.034) (Table 2).

In conventional PCR, faecal samples from 52 out of 58 dogs (89.65%) were positive for CPV and this number included all dogs positive by antigen testing (Table 2). The proportion of PCR-positive dogs was not significantly different between vaccinated and unvaccinated/unknown status dogs (*p* = 0.345).

A sequence analysis of 450 nucleotides of the VP2 gene identified CPV-2c in samples from 48 dogs (92.31%), CPV-2b in three dogs (5.77%) and CPV-2a in one dog (1.92%). There was only fair agreement, based on Cohen’s Kappa statistic of 0.34 (95% CI = 0.12–0.55), between results obtained by conventional PCR and those obtained using the antigen detection test.

The detection limit for the qPCR method using a tenfold serial dilution of the cloned plasmid and the *Ct* values for negative controls ranged from 3813 to 1278 (average, 2344) copies of the viral genome in 10 µL of DNA (represents 20 mg of faeces). The cut-off was calculated based on the mean *Ct* value for negative controls plus two times the standard deviation. The viral copy number was finally calculated per gram of faeces with a coefficient of determination (R^2^) of 0.9978 (Figure 1).

Based on the comparison of the results from serial dilutions of the positive faecal samples tested in both antigen detection and qPCR tests (Table 3), the mean limit of detection for the antigen detection kit was 8.3 × 10^8^ viral particles per gram of faeces, while it was 1.17 × 10^5^ for the qPCR. The qPCR assay was determined to be approximately 7000 times more sensitive than the antigen detection kit.

The copy number for each sample was calculated based on the mean *Ct* value for each sample compared with the standard curve. The mean *Ct* value in PCR positive samples (mean = 17.08, SD = 2.57) was significantly lower than in PCR negative samples (mean = 37.77, SD = 1.60) (*p* ≤ 0.001). The mean *Ct* value in dogs with unvaccinated or unknown vaccination status (mean = 18.20, SD = 6.08) was significantly lower than for samples from vaccinated dogs (mean = 21.69, SD = 8.03) (*p* = 0.049), suggesting reduced viral shedding in vaccinated dogs. Most vaccinated dogs (85.71%) were infected with CPV-2c, while two vaccinated dogs were infected with other genotypes. There was no association between CPV-2c and CPV vaccination status (Table 2). BLC analysis estimated the Se_AG_ and Sp_AG_ to be 72.32% and 88.26%, respectively. The Se_PCR_ and Sp_PCR_ was estimated at 97.98% and 89.20% respectively (Table 4). An optimised cut-off *Ct* value of 23.10 was chosen from the TG-ROC curve (Figure 2).

The Se_qPCR_ and Sp_qPCR_ corresponding to the chosen cut-off *Ct* were 97.53% and 95.59%, respectively (Table 3). The proportion of faecal samples that were positive when using the optimised cut-off was 89.66%. There was only a fair agreement, using Cohen’s Kappa statistic of 0.34 (95% CI = 0.12–0.55), between the proportion of dichotomised results based on the optimised cut-off on qPCR and conventional PCR and those obtained using the antigen detection test.

The application of weakly informed or good priors in BLC analysis had a low effect on the posterior median values compared to those obtained using informative priors (data not shown).

## 4. Discussion

CPV-2c was the predominant strain detected in vaccinated and unvaccinated Australian dogs affected by parvoviral enteritis between 2015 and 2019 in this study. CPV-2c is of emerging importance in Australia and findings here suggest it may be replacing CPV-2b as the predominant circulating strain in Australia [46]. Existing vaccines in Australia and elsewhere are derived from CPV-2b as well as the original CPV-2. Previous vaccination failed to prevent CPV infection in 14/17 dogs, including 12 dogs with the CPV-2c variant, one dog with the CPV-2b variant, and one dog with the 2a variant. Causes of vaccine failure beyond vaccine factors (strain; storage and administration errors) are many, however, and include conditions of high environmental viral load challenge [48], viral antigenic diversity, parasitic and bacterial enteric infections [49,50,51,52], breed [53], genetic non- responders [54] and immune incompetence [55]; these co-factors were not investigated in the current study. Vaccines do not prevent infections, but should minimize disease in immunocompetent patients in a low-virus challenge environment; in addition to evaluating the appropriateness of current vaccine strains, further investigations into mitigating co-factors associated with increased risk are recommended.

The molecular epidemiology and antigenic characterization of circulating CPVs revealed that CPV-2c is gradually replacing other CPV-2 variants in the dog population worldwide [21]. The pathogenicity and clinical features of the CPV-2c variant in dogs are different from CPV-2a and CPV-2b variants. Decaro et al., 2005, showed that enteritis caused by CPV-2c is more mucoid compared to the other CPVs that caused haemorrhagic diarrhoea In addition, the low sensitivity of antigen detection tests has been associated with high gut CPV antibody levels, which are speculated to sequester a substantial proportion of viral particles from the test [1]. Our findings have already revealed the lower virus load and less diarrhoeic features in faecal samples from CPV-2c infected dogs.

Although no statistically significant association was found between CPV genotype and vaccination status, the samples sizes for CPV-2a and CVP-2b genotypes were small, suggesting insufficient power to detect variations in vaccine protection against different genotypes. A recent Australian study found that the age at administration of the final CPV puppy vaccination was the most significant risk factor for vaccination failure and that there was no association between vaccine failure and the strain used in the vaccine’s manufacture (CPV-2 or CPV-2b) [35]. To avoid any bias due to maternal antibodies in vaccinated dogs, samples within the age range of 9–16 weeks were removed from the study.

Faecal antigen testing successfully identified all genotypes (CPV-2a, 2b and 2c), as has been reported elsewhere [56,57,58], but failed to identify a substantial proportion of CPV-2c vaccine failure cases (Table 2). In contrast, the proportion of positive dogs, as determined by PCR, was not significantly different between vaccinated and unvaccinated/unknown status clinically affected dogs. qPCR mean *Ct* values for vaccinated dogs affected by parvoviral enteritis were higher than those for unvaccinated dogs, suggesting that vaccinated dogs shed less virus during infection. As shown in Table 3, the sensitivity of the antigen detection test is heavily affected by the viral load of the tested faecal samples. This can explain the reason for the failure of the detection of an infection in clinically affected dogs with CPV-2c variants. Tests detecting faecal viral antigens represent the only patient-side tests presently available in veterinary clinics [1]; however, in dogs with clinical signs compatible with canine parvovirus disease, negative CPV faecal antigen results should be viewed with caution until they are confirmed by molecular methods [1,21]. The limit of detection of the faecal antigen test determined in this study was approximately 8.3 × 10^8^ viral particles per gram of faeces, with the quantitative PCR test almost 7000 times more sensitive than the antigen detection methods. In addition, significantly lower mean *Ct* values in PCR-positive samples compared to PCR-negative samples suggest a positive association between results from the qPCR and conventional PCR tests. Therefore, qPCR and conventional PCR offer marked improvements in sensitivity and negative predictive value over the antigen test in the study population.

The copy number for each sample was calculated based on the mean *Ct* value for each sample compared with the standard curve. The mean *Ct* value in PCR-positive samples (mean = 17.08, SD = 2.57) was significantly lower than in PCR-negative samples (mean = 37.77, SD = 1.60) (*p* ≤ 0.001). The mean *Ct* value in unvaccinated dogs or dogs with an unknown vaccination status (mean = 18.20, SD = 6.08) was significantly lower than for samples from vaccinated dogs (mean = 21.69, SD = 8.03) (*p* = 0.049), suggesting reduced viral shedding in vaccinated dogs.

Antigen-based patient-side detection kits are known to be inferior to molecular assays for the detection of CPV, which is associated with reduced faecal viral shedding during the late stages of infection and/or the early presence of high CPV antibody titres in the gut lumen that may sequestrate viral particles [25,59]. In light of the findings here, previous vaccination against CPV-2 should also be considered as a factor contributing to reduced antigen test sensitivity. The effect of the strain on the performance of the faecal antigen test was not examined in this study due to the small sample sizes for CPV-2a and CPV-2b strains. However, studies elsewhere suggest that the performance of faecal antigen tests does not differ between dogs infected with CPV-2a, 2b or 2c [56,58].

## 5. Conclusions

CPV-2c was the predominant circulating strain in both vaccinated and unvaccinated Australian dogs affected by parvoviral enteritis in this study. Faecal antigen testing was insufficiently sensitive to detect 40% of animals shedding the virus. Reduced but not eliminated viral shedding of CPV-2c in vaccinated dogs with parvoviral enteritis is of particular concern for disease transmission and epidemiology, with cases of CPV infection potentially remaining unconfirmed and inadequate disease control measures being applied. Findings in this and other studies indicate the need for ongoing molecular surveillance of circulating and emerging parvoviral strains, in addition to increasing awareness of the prevalence and characteristics of the disease in both vaccinated and unvaccinated dogs.

## Figures and Tables

**Figure 1 viruses-12-00980-f001:**
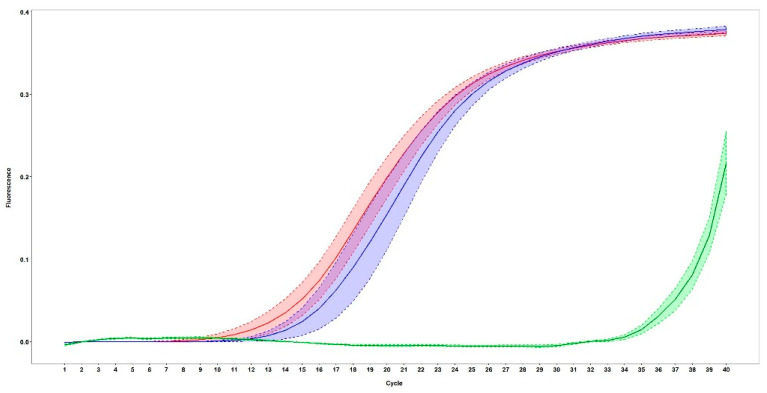
Comparison of mean and standard error of mean fluorescent signal (*ΔRn*) in qPCR for vaccinated (blue curve), unvaccinated (pink curve), and negative (green curve), samples. In total, unvaccinated PCR positive samples show lower *Ct* value suggesting higher virus load.

**Figure 2 viruses-12-00980-f002:**
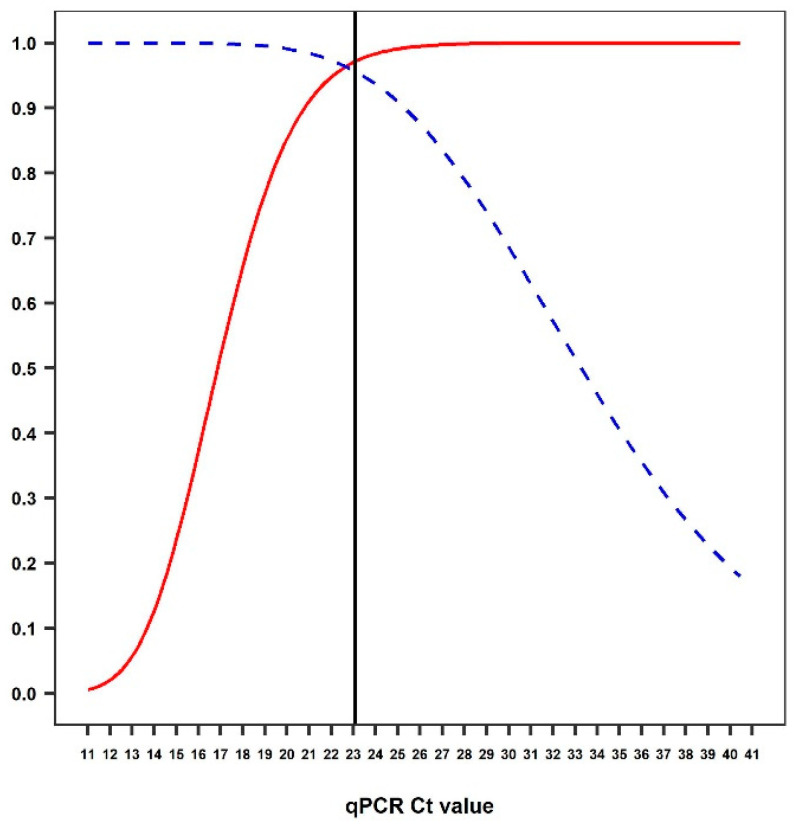
Two-graph receiver operating characteristic curve (TG-ROC) as obtained from Bayesian latent class analysis of *Ct* values for faecal samples on quantitative PCR (qPCR) test. The dashed and solid lines represent specificity and sensitivity, respectively. The cut-off was chosen at a *Ct* value of 23.10 (vertical solid line).

**Table 1 viruses-12-00980-t001:** Informative beta priors used for estimation of test sensitivity (Se) and specificity (Sp) of antigen detection and conventional PCR in study population in a Bayesian latent class model.

Parameter	Mode (%)	95% Sure Greater Than (%)	Corresponding Beta Prior Distribution (α, β)
Antigen detection sensitivity	85	50	6.25, 1.93
Antigen detection specificity	85	50	6.25, 1.93
PCR sensitivity	90	50	5.38, 1.49
PCR specificity	90	50	5.38, 1.49

**Table 2 viruses-12-00980-t002:** Summary of faecal sample testing results from antigen detection, conventional PCR, genotyping sequence analysis, and qPCR tests and their distribution in vaccinated and unvaccinated/unknown vaccination status dogs.

Tests↓	Number of Positive Samples in Different Tests (%)		Total Positive (*n* = 79)	*p*-Value
	Positive/Vaccinated	Positive/Unknown or Unvaccinated		
Antigen detection	7/32 (21.88%)	30/47 (63.83%)	37 (46.84%)	<0.001 *
Conventional PCR	14/32 (43.75%)	38/47 (80.85%)	52 (65.82%)	0.001
2c Genotype	12/14 (85.71%) ^a^	36/38 (94.74%) ^a^	48 ^a^ (92.3%)	0.29
2b Genotype	1/14 (7.14%) ^a^	2/38 (5.26%) ^a^	3 ^a^ (5.76%)	NA
2a Genotype	1 /14 (7.14%) ^a^	0/38 (0%) ^a^	1 ^a^ (1.92%)	NA
PCR Negative	3/17 (17.65%)	3/41 (7.32%)	NA	NA
qPCR mean (SD) *Ct* value ^b^	26.67 (7.91)	17.65 (7.49)	NA	<0.001 *^,c^

^a^ Testing done on faecal samples positive in PCR (*n* = 52) only; ^b^ Average *Ct* value for positive test results only; ^c^ Fisher’s (F) test *p*-value as obtained from one-way ANOVA analysis. * *p*-value significantly <0.05; not applicable (NA).

**Table 3 viruses-12-00980-t003:** Test results of antigen detection test compared with qPCR determined by testing serial dilutions of faecal samples.

Sample Code↓	Dilution→	1	10^1^	10^2^	10^3^	10^4^	10^5^	10^6^	10^7^	10^8^
A1	Ag test *	P	P	P	P	P	N	N	N	N
	Average *Ct*	12.27	15.01	17.47	20.78	24.11	26.49	28.88	30.46	32.11
	Copy # ^†^	8.85 × 10^10^	9.12 × 10^9^	8.93 × 10^8^	9.09 × 10^7^	9.09 × 10^6^	8.77 × 10^5^	8.52 × 10^4^	8.06 × 10^3^	7.85 × 10^2^
A2	Ag test *	P	P	P	P	P	P	N	N	N
	Average *Ct*	11.58	13.75	16.35	19.1	22.17	25.23	28.33	31.57	34.17
	Copy # ^†^	2.4 × 10^12^	3.5 × 10^11^	3.4 × 10^10^	2.9 × 10^9^	1.8 × 10^8^	1.2 × 10^7^	7.4 × 10^5^	4.1 × 10^4^	4.0 × 10^3^
A3	Ag test *	P	P	P	P	P	P	N	N	N
	Average *Ct*	11.32	14.25	16.88	20.58	23.13	25.97	29.71	32.88	36.83
	Copy # ^†^	3.1 × 10^12^	2.2 × 10^11^	2.1 × 10^10^	7.7 × 10^8^	7.8 × 10^6^	6.1 × 10^6^	2.1 × 10^5^	1.2 × 10^4^	3.6 × 10^3^
A4	Ag test *	P	P	P	P	N	N	N	N	N
	Average *Ct*	14.33	17.25	21.04	23.78	27.11	30.88	33.21	35.88	36.17
	Copy # ^†^	2.1 × 10^11^	1.5 × 10^10^	5.1 × 10^8^	1.8 × 10^8^	404 × 10^7^	7.6 × 10^4^	9.4 × 10^3^	8.6 × 10^2^	6.6 × 10^2^
B2	Ag test *	P	P	P	P	N	N	N	N	N
	Average *Ct*	15.25	18.57	22.31	25.41	28.82	31.83	34.14	36.74	36.81
	Copy # ^†^	9.1 × 10^10^	4.6 × 10^9^	1.6 × 10^8^	4.0 × 10^7^	4.8 × 10^5^	3.2 × 10^5^	4.1 × 10^3^	4.0 × 10^2^	3.7 × 10^2^

* Positive (P), negative (N); ^†^ copy; # = viral DNA copy number per 20 mg of faces.

**Table 4 viruses-12-00980-t004:** Test sensitivity (Se), specificity (Sp), positive predictive value (PPV), and negative predictive value (NPV) with their 95% credible intervals for antigen detection test, conventional PCR and qPCR (at optimised cut-off *Ct* value of 23.10) as obtained from Bayesian latent class analysis with informative priors.

Parameter	Antigen Detection Test	Conventional PCR	qPCR
Sensitivity (%)	72.24 (60.09–82.52)	97.98 (92.17–99.82)	97.86 (93.98–99.41)
Specificity (%)	95.39 (85.32–99.35)	96.22 (85.75–99.65)	95.51 (89.90–98.19)
Positive predictive value (PPV) (%)	96.61 (89.08–99.52)	97.92 (91.93–99.81)	97.54 (94.03–99.09)
Negative predictive value (NPV) (%)	65.37 (51.52–77.78)	96.32 (86.20–99.66)	96.09 (88.90–98.95)

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
