# Peer review of "Diagnostic Challenges in Canine Parvovirus 2c in Vaccine Failure Cases"

_viruses, 2020, doi:10.3390/v12090980_

Round 1

Reviewer 1 Report

This manuscript describes the occurrence of enteritis in dogs with and without a known vaccination history, and the predominant genotypes associated with the cases. Overall, the manuscript presents some worthwhile data, but the conclusions are a bit misleading.

A few general comments, since other reviewers have done a good job with clarifications and editing.

1) Good to clarify that vaccines do not prevent infections, but do minimize disease in an immunocompetent dog, in a low virus challenge environment, and in outbred dogs.

2) PCR detects CPV in low level shedding conditions, whereas CPV Ag ELISA is used as a dog side to aid in the diagnosis of CPV enteritis. The Ag ELISA will detect CPV in the 4 peak Ag shedding days over the course of the 10-12 days of clinical disease.

3)Discussion. Reasons for viral enteritis in well vaccinated dogs are as noted above: overchallenge conditions; multicomponent enteric infections with viruses and bacteria; breed predilection, which were not discussed in this manuscript.

4) Recent papers for authors to refer to are: Decaro, Buonavoglia, Barrs, 2020.108760. Vet. Microbiology ; Kelman, Norris, Barrs, Ward, 2020.13002, Australian Vet J.

Author Response

Response to reviewers for manuscript ID: viruses-906915

Title: Diagnostic Challenge in Canine Parvovirus 2c in vaccine failure cases.

Please find our responses in Italic to the reviewers’ comments:

Reviewer 1:

Comments and Suggestions for Authors

This manuscript describes the occurrence of enteritis in dogs with and without a known vaccination history, and the predominant genotypes associated with the cases. Overall, the manuscript presents some worthwhile data, but the conclusions are a bit misleading.

  • Thanks for the comments on our paper, we have tried to revised the discussion of the manuscript and provide more description on the significance of our work compare to the published data.

A few general comments, since other reviewers have done a good job with clarifications and editing.

1) Good to clarify that vaccines do not prevent infections, but do minimize disease in an immunocompetent dog, in a low virus challenge environment, and in outbred dogs.

  • We have provided some references and more clarification at the introduction regarding the role of the CPV vaccines in infection

2) PCR detects CPV in low level shedding conditions, whereas CPV Ag ELISA is used as a dog side to aid in the diagnosis of CPV enteritis. The Ag ELISA will detect CPV in the 4 peak Ag shedding days over the course of the 10-12 days of clinical disease.

  • The accuracy of the tests in different steps of the infection is one of the biggest challenges in all diagnostic labs especially for the infection diseases. We have tried to concentrate on the values of the diagnostic tests in different CPV genotypes. We have treated our samples as blind diagnostic samples and tried to get results that are more accurate in different genotypes. As it has discussed at the manuscript, CPV-2c is responsible for diagnostic failure in In-clinic screening tests.

3)Discussion. Reasons for viral enteritis in well vaccinated dogs are as noted above: overchallenge conditions; multicomponent enteric infections with viruses and bacteria; breed predilection, which were not discussed in this manuscript.

  • We have added a paragraph to the discussion and discuss about the other enteric pathogens, we still heavily concentrate on the role of different CPV genotypes on vaccine failure cases with or without diagnostic challenges.

4) Recent papers for authors to refer to are: Decaro, Buonavoglia, Barrs, 2020.108760. Vet. Microbiology ; Kelman, Norris, Barrs, Ward, 2020.13002, Australian Vet J.

  • The recent papers have added to the reference list and discussed at the discussion part of the paper. We believe these two recent published papers have increased the value of our work when we compared our results with the results in these papers.

Reviewer 2 Report

Unfortunately, no traceability of the corrections is present. Despite the explanations given in the reply to the auditors, I do not see the corrections included in the text (for example the fact that the origin of the samples is from 5 different places: many data from Australia have been recently published on CPV-2 and FPV).

Some mistakes are still present (CPV/PCV, number close the measure of units).

The main interesting point is that viral load in vaccinated animals is lower than in non-vaccinated ones.

Author Response

Response to reviewers for manuscript ID: viruses-906915

Title: Diagnostic Challenge in Canine Parvovirus 2c in vaccine failure cases.

Please find our responses in Italic to the reviewers’ comments:

Reviewer 2:

Comments and Suggestions for Authors

Unfortunately, no traceability of the corrections is present. Despite the explanations given in the reply to the auditors, I do not see the corrections included in the text (for example the fact that the origin of the samples is from 5 different places: many data from Australia have been recently published on CPV-2 and FPV).

  • Thanks for the comments, we have uploaded the previous version of the manuscript with the tack changes and also a supplementary file contains all details from the collected samples has uploaded via the journal submission system.

Some mistakes are still present (CPV/PCV, number close the measure of units).

  • All of the corrections have made in the manuscript and are traceable through the older version with tack changes.

 The main interesting point is that viral load in vaccinated animals is lower than in non-vaccinated ones.

  • Thanks again for the valuable comments; we believe that both reviewers’ comments helped us to improve the quality of our manuscript.

This manuscript is a resubmission of an earlier submission. The following is a list of the peer review reports and author responses from that submission.

Round 1

Reviewer 1 Report

The novelty of the results of the study is quite limited.

It is interesting the hypothesis that vaccination is partially protective; the current opinion is that the vaccine is hindered by maternal antibodies and the vaccine in this situation does not work.

The main problem is that the aim of the study stated in the introduction does not fit with the workflow of the study. This study is not designed to perform a prevalence study and the dogs come from a single site. It is too hard to say that "CPV-2c was the predominant variety...in the Australian dog ...". In addition, new varieties of strains are emerging worldwide, still classified as CPV-2a, 2b and 2c.

On the other hand, if the authors want to study the performance of a novel qPCR test, they should have an excellent description of the clinical cases. There is a poor standardization of cases: parvovirus infection can cause a wide range of clinical signs. On the other hand, gastrointestinal diseases are one of the most frequent reasons for a first visit to a veterinary clinic. For example, are you sure that the six negative cases are really negative? What diagnosis did they have in those cases?

The paragraph at lines 302-309 ("Patient side based on antigen ...") must be merged with one at lines 278-295 ("Fecal antigen ..."): the most recognized reason of the low sensitivity of the antigen test is the presence of antibodies in the gut. The sensitivity of these tests reported in some studies is 50% (Buonavoglia, Decaro 2012; Veil 2014). The authors should have used this percentage in the beta priors data.

No time between vaccination and onset of clinical signs has been reported. Vaccinal DNA may be present in 5 month old puppies if recently vaccinated. Are the authors sure to rule out this possibility? Generally a real-time PCR with DNA probes is used as a gold standard in the case of recent vaccinations.

The comparison between dead and surviving animals could have been interesting to understand the effect of vaccination on these cases. In fact, puppies vaccinated within a few days can die for parvovirus infection.

References 1 and 31 are the same.

There are mistakes in the presentation of the missing references 54-61.

Reviewer 2 Report

In the manuscript "Diagnostic Challenge in Canine Parvovirus 2c in 2 Vaccine Failure Cases" characterization of CPV strains isolated from dogs with gastroenteritis and diagnostic efficiency of some molecular biology tests compared with in-clinic tests are reported.

The manuscript lacks completely of originality as the topics have already been widely discussed elsewhere for several decades. Some main shortcomings of the study are highlighted below:

  1. It has been repeatedly demonstrated that "in-clinic" tests are poor sensitive. Where is the original result in the study?
  2. It seems somewhat laborious carrying out PCR test and then sequencing the amplicon to characterize the isolated strains. Molecular methods that quantify and characterize the virus at the same time have been widely used for a long time with considerable time savings.
  3. Epidemiological information on the diffusion of the variants is now lacking of any interest, since such studies have been already carried out worldwide.
  4. Some considerations appear arbitrary: "qPCR mean Ct values ​​for vaccinated dogs affected by parvoviral enteritis were higher than those for 2 unvaccinated dogs, suggesting that vaccinated dogs shed less virus during infection." Noteworthy, the amount of virus eliminated in the stools during infection depends on multiple factors:
  5. the moment of infection in which the stool sample is collected: during the progression of the infection viral titre decreases;
  6. titre of maternal antibodies in puppies can modulate the excretion of the virus in the stool in terms of titre and duration of the excretion.

According to what specified, the manuscript cannot be published in the prestigious journal "Viruses".

We recommend trying to publish the manuscript to a low impact journal.

Reviewer 3 Report

This manuscript describes a good study comparing the occurrence of CPV associated disease in clinically affected dogs with enteritis. The authors compare the CPV Ag ELISA with PCR in various formats, and in vaccinated vs unvaccinated dogs.

There is a tendency (bias) to: overlook the CPV carrier population (low shedders) in the dog population; the fact that vaccination does not prevent infection/re-infection, but is used in practice to minimize disease; and that vaccination may be overcome by "over challenge" with CPV in the dog population. The discussion should a least mention these to be thorough and let the reader know of other possibilities and the limits of this research.

Other comments for the authors to consider are as follows:

Page 1, line 40. ...such as mink and foxes.

Page 2, line 59. ...and possibly adaptation of the virus to the host via genetic selection.

Page 2, line 65. ...controlling CPV disease and increasing population immunity in dog environments.

Page 2, line 77. ...CPV-2 isolates of clinical cases..

Page 2, line 84. Delete last incomplete sentence.

Page 2, line 87. What was the result of the canine coronavirus data from the Ag detection assay? Clarify for reader.

Page 3, lines 139-144. This terminology usually refers to positive and negative predictive value for clinical disease. But this needs a negative cohort for analysis. Clarify for reader.

Page 6, Table 2. This data from clinically affected dogs only. Difficult to determine positive and negative predictive value without the negative cohort (see above).

Page 6, lines 220-222. The PCR was shown to be very sensitive  compared to the Ag ELISA. To determine the positive negative predictive value would need to identify what is the percent of subclinical dogs that are shedding CPV in the feces. Clarify for the reader.

Page 9, line 302. Ag based ELISA for CPV has been demonstrated to have high positive predictive value for clinical disease during the 4 peak antigen shedding days. It will not generally detect CPV carrier dogs, which are low shedders. Clarify for reader from the authors conclusions.